# Reviewing the Source, Physiological Characteristics, and Aroma Production Mechanisms of Aroma-Producing Yeasts

**DOI:** 10.3390/foods12183501

**Published:** 2023-09-20

**Authors:** Li Chen, Ke Li, Huitai Chen, Zongjun Li

**Affiliations:** 1College of Food Science and Technology, Hunan Agricultural University, Changsha 410128, China; lichen99199@163.com (L.C.); leeke14@163.com (K.L.); 2Hunan Guoyuan Liquor Industry Co., Ltd., Yueyang 414000, China; 18673187905@163.com

**Keywords:** aroma-producing yeast, microorganism, aromatic compounds, terpenoids

## Abstract

Flavor is an essential element of food quality. Flavor can be improved by adding flavoring substances or via microbial fermentation to impart aroma. Aroma-producing yeasts are a group of microorganisms that can produce aroma compounds, providing a strong aroma to foods and thus playing a great role in the modern fermentation industry. The physiological characteristics of aroma-producing yeast, including alcohol tolerance, acid tolerance, and salt tolerance, are introduced in this article, beginning with their origins and biological properties. The main mechanism of aroma-producing yeast is then analyzed based on its physiological roles in the fermentation process. Functional enzymes such as proteases, lipases, and glycosidase are released by yeast during the fermentation process. Sugars, fats, and proteins in the environment can be degraded by these enzymes via pathways such as glycolysis, methoxylation, the Ehrlich pathway, and esterification, resulting in the production of various aromatic esters (such as ethyl acetate and ethyl caproate), alcohols (such as phenethyl alcohol), and terpenes (such as monoterpenes, sesquiterpenes, and squalene). Furthermore, yeast cells can serve as cell synthesis factories, wherein specific synthesis pathways can be introduced into cells using synthetic biology techniques to achieve high-throughput production. In addition, the applications of aroma yeast in the food, pharmaceutical, and cosmetic industries are summarized, and the future development trends of aroma yeasts are discussed to provide a theoretical basis for their application in the food fermentation industry.

## 1. Introduction

In the three-domain classification system, yeast belongs to the Fungi kingdom, alongside the Plantae and Animalia kingdoms. It is a single-celled fungus with a different cellular structure and function from other fungi, with over 1500 species. Yeast is widely used, for example, in the food industry where metabolic products produced during the fermentation process, such as *Saccharomyces cerevisiae*, *Saccharomyces exiguus*, and *Kluyveromyces lactis*, can be used to impart specific flavors and textures to food; and in the cosmetics industry where yeast extract and yeast fermentation products are commonly used in skincare and makeup products, with moisturizing, antioxidant, and skin reparation effects. *Saccharomyces cerevisiae* can break down carbon sources into CO2 and alcohol, and thus complete alcoholic fermentation [1], while some non-*Saccharomyces cerevisiae* are initially considered contaminated due to their low alcohol tolerance and intolerance to acidity, thus producing undesirable odors, e.g., film-forming yeasts and *Pichia anomala*, which may contaminate wine and wine production machines, respectively [2]. However, with the growth of the fermentation industry in recent years, it has been found that via appropriate screening methods, non-*Saccharomyces cerevisiae* can be used in fermentation to produce specific aromas, which bring special qualities to the wine after fermentation, and non-*Saccharomyces cerevisiae* are gradually emerging as a source of aromatic compounds in wine [3,4]. *Aroma-producing yeast* is a class of strains that can produce alcohols, aldehydes, organic acids, and furan compounds with aromatic metabolites via fermentation. Its aroma is dominated by esters and alcohols, mainly ethyl acetate [5], and is similar to the apple aroma [6]. It plays a significant role in the modern process of brewing white wine. In addition, it is also widely used in the production of drinking wines, soy sauce, spices, and some fermented pasta products, with the ability to improve the aroma of finished products and improve the stability of the product’s quality. In addition to traditional *Saccharomyces cerevisiae*, non-*Saccharomyces cerevisiae* is also applied in the fermentation industry, but unlike *Saccharomyces cerevisiae*, which can produce alcohol, its primary role is to generate a large number of volatile aromatic compounds during the fermentation process so as to enhance the flavor of the product; this means that fermentation can be used to produce various low-alcohol beverages with different aromatic types. Aroma-producing yeast can affect primary and secondary aromas by producing enzymes and metabolites, respectively. Most primary aromatic compounds exist in a free or bound form and can be hydrolyzed via the action of yeast during fermentation. Of particular importance are the aroma precursors linked to sugar molecules [7]. At present, the main aroma-producing yeast in China are *Hansenula* [8], *Candida*, *Pichia anomalous* [9], etc. The selection and breeding processes of aroma-producing yeast are not very different from those of other yeast genera and only slightly differ in fermentation temperature and pH [10]. Fermentation at lower temperatures (15–18 °C) increases the production of volatile compounds (esters and alcohol) [11]. These volatile compounds are the main source of aroma in fermented products, but there is a lack of research on aroma-producing yeasts and their aroma-production mechanisms. Therefore, the purpose of this review is to provide a systematic introduction to this topic and to summarize the fragrance-production mechanism.

## 2. Source of Aroma-Producing Yeast

Aroma-producing yeast is widely present in nature and is widely distributed on fruits and their epidermis. Some aroma-producing yeast and their sources are shown in Table 1. In the early days of wine-making, yeast was derived mainly from vineyards and grape skins, with small amounts also present on the surfaces and contact surfaces of wine-making equipment [12]. Yeasts from different regions also have unique characteristics. Wang et al. [13] fermented small ginseng and grape juice via sequential inoculation with native Yantai yeasts H30 and YT 13, and the results showed that the lactic acid contents of the natural wines fermented with H30 and YT 13, respectively, were higher than that of the control group fermented with ferulic acid in a single fermentation, while the mixed fermentation group was also found to contain butyrolactone and propionic acid anthocyanins, which increased the complexity of the aroma of small ginseng wine, and provided a high-quality yeast resource for the production of white wine with unique regional characteristics. Liu et al. [14] extracted four strains of aromatic yeast from citrus peels from Sichuan after analyzing and identifying the morphological, physiological, and biochemical characteristics and ITS sequence features, enabling them to produce a strong aroma. Lu et al. [15] isolated a strain of raw aroma yeast, X-5, from kiwi fruit, branches, leaves, and the soil under the trees that was suitable for kiwifruit fermentation, and this resulted in a strong aroma, clear juice, and high sensory quality; it could thus be applied as a special strain for kiwifruit wine fermentation. Tang et al. [16] screened four strains of salt-tolerant and strong aroma-producing bacteria from pickled vegetables in southern Sichuan, providing theoretical support for the preparation of pickled fermentation agents.

## 3. Biological Properties of Aroma-Producing Yeast

There are many types of aroma-producing yeast, and the properties and functions of each genus are different. For example, during the fermentation of plum wine with high acidity, *Hansenula* can reduce the content of flavanols, thus reducing the astringency and bitterness of the wine. It can also produce a large amount of isobutyl acetate, ethyl butyrate, and ethyl caproate, which bring strong fruit flavors to the wine. In the same study, *Pichia* and *Issatchenkia orientalis* were found to have high malic and citric acid degradability, thus reducing the acidity and astringency of the wine [28] and improving its taste.

### 3.1. Pichia

The cells of *Pichia* are characterized by short oval- to cylindrical-shaped colonies, which can use methanol, glucose, glycerol, sorbitol, or ethanol [29,30,31] as a carbon source to produce organic acids and esters during the fermentation process, giving the product a strong flavor. They also have high-density fermentation properties and high-performance expression activity [32]. Kuang et al. [33] obtained a strain of *Pichia kluyveri* with good aroma-production performance during the isolation of coffee-fruit wine aroma yeasts, and found that the fermentation broth contained a large number of aroma components, which brought about pleasant fruit and floral aromas. They proved via experiments that the aroma components were mainly esters (32.34%), alcohols (26.26%), acids (34.57%), and other volatile compounds, thus contributing to the theoretical basis for the fermentation of fruit wines and the improvement of the aroma of coffee country wines using *Pichia kluyveri*. Liu et al. [34] isolated two *Pichia kudriavzevii* strains from the naturally fermented juice and peels of mulberry, which were more alcohol-tolerant than *Hansenula*, and produced two to six more aroma components after fermentation than *Hansenula.* Aroma-producing yeast can enrich the quality of fruit wine and bring a strong floral aroma. Wang et al. [35] obtained a strain of high-β-phenyl ethanol-producing *Pichia* from high-temperature stacked wine grains of Jiahu via isolation and purification screening and analyzed the volatile flavor substances of this yeast after fermentation via gas chromatography. The results showed that the strain could synthesize a variety of compounds that contribute to the flavor of white wine, such as phenyl ethanol and phenethyl acetate, which are important guides for the fermentation and production of white wine. Β-phenyl ethanol is an important flavor substance in traditional fermented foods, such as white wine, low-alcohol beverages, soy sauce, and bread, and has a soft, light rose aroma. Fu et al. [25] used the spread plate method to obtain a strain of *Pichia guilliermondii*, giving a high yield of β-phenyl ethanol from old white dry aromatic wine malt and used its fermentation aroma-producing characteristics to produce about 20 aroma compounds in the culture medium.

### 3.2. Hansenula

*Hansenula* is a typical methylotrophic microorganism that can use methanol as the sole carbon source for high-density growth. Its methanol utilization pathway is similar to that of *Pichia pastoris*, which is also carried out in peroxisomes. The difference is that *Hansenula* only has one methanol oxygenase (Mox), and methanol regulation is not as strict as *Pichia pastoris*. It can also achieve a certain level of Mox expression in low concentrations of glycerol and glucose; the commonly used fermentation carbon sources are mainly glycerol and methanol [36]. In multilateral budding and fermentation, *Hansenula* yields a white mold on the liquid surface, a murky culture solution, robust fermentation and esterification power, and the ability to accumulate tryptophan. It is primarily round, oval-, or sausage-shaped. Tang et al. [37] used bioinformatics to analyze the functions of differentially expressed genes in *Hansenula anomala* at five fermentation time points (0, 24, 48, 72, and 96 h), and the results showed that *Hansenula anomala* had strong metabolic and genetic information processing activities at 96 h. Sequential inoculation of non-*Saccharomyces cerevisiae* and *Saccharomyces cerevisiae* can cause changes in the chemical composition of wine, thereby affecting its sensory characteristics. Izquierdo et al. [38] improved the aroma and sensory quality of white wine by using abnormal *Hansenula anomala* and *Saccharomyces cerevisiae*, resulting in greater aromatic complexity. Ai et al. [39] isolated two strains of *Hansenula anomala* and *Brettanomyces Claussenii* from naturally fermented citrus juices, with good aroma production results. Despite both strains’ poor alcohol tolerance, they can both tolerate high concentrations of salt and sugar and are suitable for use in juice concentrates or the aromatization of low-alcohol fruit wines.

### 3.3. Zygosaccharomyces

*Zygosaccharomyces* has high sugar, salt, and alcohol tolerance, and it plays an active role in the fermentation of some foods and beverages, such as balsamic vinegar [40] and soy sauce [41]. *Saccharomyces rouxii* is also used as the main yeast in the production of Pixian bean paste, which can tolerate up to a 20% sodium chloride mass fraction, and it has strong alcohol- and aroma-production capacities [42]. It also participates in the formation of flavor substances in the middle and later stages of soy sauce fermentation and can produce small-molecule alcohols such as ethanol, isoamyl alcohol, glycerol, arabinoxylan, and other sugar alcohols [43]. It also has excellent ester production capacity [44]. Li et al. [45] used five non-enological yeasts in fermentation to improve the functional properties and flavor characteristics of kiwifruit wine, among which *Zygosaccharomyces rouxii* and *Zygosaccharomyces bailii* increased the ethyl ester contents and performed better in terms of aroma production than the enological yeasts. In addition, *Zygosaccharomyces bailii* is very resistant to ethanol and some weakly acidic preservatives (e.g., sorbic acid), and it is also considered a serious spoilage yeast in the food industry, causing significant losses to breweries, juice production plants, etc. [46]; however, it can be controlled via the use of antimicrobial peptides such as lactoferrin B, dimethyl dicarbonate (DMDC), or non-thermal sterilization techniques [47]. In balsamic vinegar, *Zygosaccharomyces rouxii* is responsible for the conversion of high concentrations of sugar to ethanol, with a preference for fructose over glucose (a trait called fructophily); it also releases important flavor substances such as heterocyclic alcohols and 4-hydroxy furanone derivatives, which promote aromatic and smoky flavors in soy sauce [48], and has been successfully used in the manufacturing of low-ethanol products.

### 3.4. Candida

*Candida* is an acid-producing, cold-tolerant, and highly permeable strain [49], which has a rich genetic diversity and unique metabolic properties that provide diverse organoleptic capacities over a single *Saccharomyces cerevisiae* fermentation, and they have been discussed as a potential fermenter in industrial mixed fermentations [50,51]. Cai et al. [52] used Candida for aroma enhancement in the brewing of a new type of biopeptide yellow wine, and the results showed that the addition of pseudo filament yeast culture at 2–3% during compound fermentation could significantly enhance the aroma of yellow wine; the wine also had a more mellow taste and better overall quality. In addition, *Candida* can produce antifungal compounds with protein properties during fermentation, and these can affect the growth of *Bruttanomyces bruxellensis* without inhibiting the growth of *Saccharomyces cerevisiae*, reducing the spoilage brought about by the microorganism *Bruttanomyces bruxellensis* during brewing [53].

### 3.5. Other Yeasts

In the fermentation industry, there are many species of aroma-producing yeasts. Besides their aroma-enhancing effects, *Zygosaccharomyces rouxii* [54,55] and *Torulopsis* have also shown good salt tolerance, which is mainly exploited in the production of soy sauce to impart a special flavor after fermentation. *Torulopsis* is a post-mature yeast that imparts an aroma to the main body of the soy sauce by synthesizing phenolic substances such as 4-Ethylguaiacol and 4-Ethylguaiacol [56]. The combined use of the atmospheric pressure room temperature plasma (ARTP) and adaptive laboratory evolution (ALE) techniques by Li [57] significantly improved NaCl tolerance in yeast via increased intracellular K+ accumulation and cytoplasmic Na+ removal, as well as promoting glycerol production, enhancing the cell membrane and cell wall integrity, increasing the alcohol, acid, and aldehyde production, and increasing the presence of ester species. Zhang et al. [58] added *Wickerhamomyces anomalus* to fermented steamed buns, and the alcohol and ester contents increased by 11.27% and 8.85%, respectively, compared with buns without raw aromatic yeast, providing a theoretical basis for increasing the flavor substances in industrially produced buns.

## 4. Physiological Characteristics of Aroma-Producing Yeast

Yeast is a typical eukaryotic cell, with a defined variety of organelles, plasma membranes, etc. It is generally round or oval-shaped, with stable genetic material, and most of the cells are resistant to various stresses, such as high salt osmotic stress and high alcohol concentration stress, as well as high-acid and high-temperature environments. When yeast is in an unfavorable environment, the normal state of the cell will be maintained via the secretion of some resistance products or by regulating the genes.

### 4.1. Alcohol Resistance Characteristics

As a metabolic product of yeast, ethanol promotes fermentation and product flavor formation. However, with increasing alcohol concentrations, it begins to have an inhibitory effect on yeast, and at high concentrations, ethanol disrupts the protein conformation and activity of key glycolytic enzymes, leading to the protein denaturation and dysfunction of the enzymes [59]. In addition, the osmotically active substances (mainly glucose and fructose) produced by high-glucose stress-tolerant grapes containing yeast can also enhance alcohol tolerance in yeast, whereas fructose acts as an inhibitor, or increases ethanol toxicity in yeast [60]. Yeast cells develop a tolerance to alcohol under various biotic and abiotic stresses, and altering the cytoplasmic composition is one of its defense mechanisms. Archana et al. [61] compared the alcohol tolerance and cellular fatty acid compositions of *Saccharomyces cerevisiae* and non-*Saccharomyces cerevisiae*. It was found that *Saccharomyces cerevisiae* tolerated 15% alcohol, while non-*Saccharomyces cerevisiae* tolerated only 10% alcohol, and it was found that under ethanol stress, the proportion of monounsaturated fatty acids increased and that of polyunsaturated fatty acids decreased in *Saccharomyces cerevisiae*, while the content of monounsaturated fatty acids decreased in non-*Saccharomyces cerevisiae*, which could partly explain its inability to tolerate more than 10% alcohol. In addition, under mixed *Saccharomyces cerevisiae* and non-*Saccharomyces cerevisiae* culture conditions, non-*Saccharomyces cerevisiae* can contribute to ethanol tolerance in *Saccharomyces cerevisiae* by forming appropriate metabolites, such as glycerol and alanine, and/or by altering the intracellular amino acid pool [62]. Mutant strains with at least partly improved fermentation performance and alcohol tolerance can also be obtained via ultraviolet (UV) irradiation and diethyl sulfate (DES) mutagenesis of non-enological yeasts in the production of wine, providing theoretical support for the broadening of industrial applications of non-enological yeasts that improve the flavor of wine. 

### 4.2. Acid-Resistant Characteristics

The various acids produced during fermentation not only combine with various alcohols to form esters with strong aromas but also have a direct impact on the quality of the wine. Heat-tolerant non-*Saccharomyces cerevisiae* can improve wine quality by producing acetic acid to change the acidity [63]. When acetic acid accumulates to toxic levels in the fermentation environment, it inhibits yeast growth, leading to fermentation arrest. In addition, it acts as an uncoupling agent, disrupting the proton motility in yeast. The haa1p gene of *Saccharomyces cerevisiae* responds to the elevated acetic acid concentrations by regulating genes involved in protein kinases, multidrug resistance transporter proteins, and proteins involved in lipid metabolism, with the expression of SAP30 and HRK1 genes acting most strongly in protecting yeast cells from acetic acid stress [64]. Palma et al. [65] also found that the Z. bailii transcription factor (ZbHaa1), homologous to *Saccharomyces cerevisiae* Haa1, is a bifunctional transcription factor that is essential to yeast response and tolerance under acetic acid and copper stress. Under other external oxidative stress conditions, the biofilm acts as a first layer of defense, and non-*Saccharomyces cerevisiae* can achieve resistance to external stresses by decreasing its polyunsaturated fatty acid and squalene contents, increasing its monounsaturated fatty acid content, and adjusting its lipid composition [66]. Furthermore, acetic acid causes ATP depletion, leading to energy depletion in cells, but acetic acid-stressed cells contain many energy-depleting defense mechanisms such as glycolysis, TCA cycle, and oxidative phosphorylation stimulation that contribute to the increase in ATP synthesis under acetic acid stress, and consequently help to prevent energy depletion in response to acetic acid and thus maintain cellular normality [64].

### 4.3. Salt Tolerance

Yeast must have high osmotic tolerance in order to survive during the fermentation of high-salt products such as soy and other sauces, so a high osmotic glycerol (HOG) response pathway is found in some high-osmotic tolerance yeasts [67], which maintains the homeostasis in osmotic pressure across the plasma membrane by synthesizing and storing glycerol intracellularly. This enables it to maintain the intracellular water content and thus the proper functioning of the cell. The salt-tolerant *Zygosaccharomyces rouxii*, which is used in making the traditional condiment miso and other sauces in Japan, also responds to high salt stress by accumulating glycerol [68]. Two genes of GPDH (ZrGPD1 and ZrGPD2) and GDH (ZrGCY1 and ZrGCY2), which encode glycerol-3-phosphate dehydrogenase and glycerol dehydrogenase, respectively, were isolated from *Saccharomyces cerevisiae*, and it was found that GPDH and GDH may play key roles in maintaining the osmotic balance of cell membranes by synthesizing or isomerizing glycerol as a compatible solute under salt stress [69]. In Wang et al. [70]’s study, by comparing the physiological and transcriptional characteristics of *Zygosaccharomyces rouxii* under salt stress, it was found that they both underwent many changes, among which the physiological changes included the accumulation of intracellular glycerol and glucan, changes in the intracellular contents of certain amino acids, etc. Furthermore, with changes at the transcriptional level, the genes involved in carbohydrate transport and metabolism, amino acid metabolism, etc., were found at 6%. The expressions of genes involved in carbohydrate transport and metabolism, and amino acid metabolism, at 6% and 12% NaCl concentration levels were significantly different from those at 0% NaCl concentration, and these physiological and transcriptome changes were well correlated, which lays the foundation for further investigations into the salt tolerance mechanism of *Zygosaccharomyces rouxii* and the improvement of its salt tolerance.

## 5. Interaction between Aroma-Producing Yeast and Other Microorganisms

Non-*Saccharomyces cerevisiae* cultures that are about 79% pure are currently available on the market, while the rest are mostly mixed fermenter combinations, mainly comprising *Saccharomyces cerevisiae* and non-*Saccharomyces cerevisiae*, and these mixed commercial *Saccharomyces cerevisiae* show great biocompatibility in addition to producing pleasant aromatic substances in the wine-making process [71]. Therefore, in the brewing process, *Saccharomyces cerevisiae* and non-*Saccharomyces cerevisiae* are generally used as part of a mixed fermentation method. Most non-*Saccharomyces cerevisiae* cannot produce alcohol, so *Saccharomyces cerevisiae* must be added to complete the brewing process, and at the same time, non-*Saccharomyces cerevisiae* produces a variety of extracellular enzymes that react with aroma precursors to produce esters, terpenes, acids, etc., which are beneficial to the aroma and taste of the wine. To improve the flavor and taste of kiwifruit wine with a single *Saccharomyces cerevisiae*, Jin et al. [72] conducted a mixed fermentation with *Hanseniaspora* and Angie’s fruit wine yeast, with *Hanseniaspora* inoculation at 4% 36 h in advance and a 1:1.5 (volume ratio) inoculation ratio of *Saccharomyces cerevisiae* to non-*Saccharomyces cerevisiae*. They obtained a kiwifruit wine with a strong aroma and a mellow mouthfeel, accompanied by floral and fruit aromas. The antioxidant capacity of mixed bacterial fermentation was stronger than that of mono-bacterial fermentation. Wang et al. [73] increased the aroma substances and fermentation rate of cider via the sequential mixed bacterial fermentation of *Hanseniaspora valbyensis* and *Saccharomyces cerevisiae*, and the results show that the fermentation time of mixed yeast fermentation was 31 d at 15 °C. The total ester concentration in the fermented cider was 1.60 times higher than that of the brewer’s mother-fermented cider, and the total residual sugar content was lower than 4.0 g/L, which addressed the shortcomings of the *Saccharomyces cerevisiae*-fermented cider that had an insufficient ester aroma. Mixed fermentation with *Saccharomyces cerevisiae* and non-*Saccharomyces cerevisiae* can be used to change the microbial composition and physiological activity of the fermentation broth by changing the inoculation method and acting on the yeast to produce different aroma compounds, ultimately enhancing the quality of the fermentation product [74]. Mixed fermentation reduces the production of two undesirable odor compounds derived from *Saccharomyces cerevisiae*, octanoic acid and n-decanoic acid, and also reduces the production of acetic acid by *Issatchenkia orientalis*, while the inoculation of *Issatchenkia orientalis* increases the content of esters such as ethyl butyrate, ethyl isovalerate, and ethyl caproate in the fermentation broth, which enhances the fruity aroma of the wine [75]. Inoculation with *Saccharomyces cerevisiae* Y7#09, *Clavispora lusitaniae* YX3307, or a mixture of both with *Hyphopichia burtoniiF* F12507 during the fermentation of macerated wine can alter the fungal community’s structure, and increase the content of ethyl hexanoate [76]. The ester production screening of 38 yeast strains cultured on synthetic microbial media demonstrated that *Hansenula* 1471 was able to consume more than 90% of the initial sugar and produce acetic acid, medium-chain fatty acids, and ethyl acetate during wine, making it a well-mixed fermenting agent [77]. The interaction between *Saccharomyces cerevisiae* and non-*Saccharomyces cerevisiae* leads to poor fermentation; therefore, fermentation is enhanced by the use of an aroma-producing yeast that is closely related to *Saccharomyces cerevisiae*, due to the over-expression of genes and the partial alleviation of NCR (nitrogen catabolite repression) in the mixed fermentation process involving the glucan fermentation pathway [78].

## 6. Mechanisms of Yeast Aroma

During mixed fermentation using aroma-producing yeast and *Saccharomyces cerevisiae*, the aroma-producing yeast participates in the first stage of fermentation as a starter yeast. It contains a variety of extracellular enzymes, such as pectinase, β-glucosidase, lipase, protease, etc. These enzymes can hydrolyze raw materials during the fermentation process, and a series of biochemical reactions occur that produce a variety of pleasant aromatic substances and add aromatic complexity to the wine. Lipases hydrolyze fats to produce fatty acids and glycerol, proteases hydrolyze proteins to produce free peptides and amino acids [79], and important intermediates such as pyruvate and a-keto acids are produced through the gluconeogenesis pathway. These intermediate metabolites generate aldehydes under the action of related enzymes, which finally produce more acids and alcohols via oxidation and reduction reactions. Among them, the presence of glycerol and a higher alcohol content are important components contributing to the sweetness and mellowness of the wine [80]. Amino acids can generate more alcohols with one less carbon atom via decarboxylation and deamination; these more abundant alcohols and acids undergo dehydration condensation reactions either directly or indirectly to produce various esters, especially ethyl acetate, ethyl caproate, ethyl lactate, etc., which are extremely important aromatic substances produced during the fermentation process, as shown in Figure 1.

### 6.1. Formation of Esters

When the synthesis of unsaturated fatty acids and sterols stops, and the amount of acetyl-CoA reaches its maximum, a fast rate of condensation with alcohols follows, which forms esters [81]. There are two main types of esters in wine. One is acetate, which is produced by the reaction of acetyl-CoA with the abundant alcohols resulting from the degradation of amino acids or carbohydrates, such as those related to a fruity flavor (ethyl butyrate, ethyl caproate, and ethyl 2-methyl butyrate), an apple flavor (ethyl isovalerate), and a banana flavor (isoamyl acetate) [82]. The other is ethyl ester, which is mainly formed during yeast fermentation via enzymatic grape precursors and the methanolysis of acyl coenzymes formed during the degradation of fatty acid synthesis, and their concentrations are influenced by yeast strain, fermentation temperature, aeration level, and sugar content [83]. 

Esters involved in fermentation can impart a rich fruit aroma. The single ester aroma threshold is not high, and the cumulative effect of multiple ester aromas gives the product its final aromatic quality. This means that the use of different raw materials, strain origins, and yeast species produces wines with different aromas [84]. In addition, there are non-aromatic precursors in grapes that release the enzymes that produce aromatic compounds [85] and acetate esters, such as isoamyl acetate (banana-like aroma) and 2-phenethyl acetate (fruity and floral aroma) [77]. The ability of *Saccharomyces cerevisiae* to convert organic acids into their corresponding esters is related to the presence of esterases within its cells [86].
$$\mathrm{RCOSCoA}+C_{2}H_{5}\mathrm{OH}\overset{\mathrm{esterase}}{\Leftrightarrow }\mathrm{RCOO}C_{2}H_{5}=\mathrm{SHCoA}$$

### 6.2. Formation of Terpenoids

Terpenes are also important components of aroma compounds and exist in free and glycosylated forms of grapes. The terpenoid biosynthesis pathway, also known as the isoprenoid biosynthesis pathway, consists of the mevalonate pathway (MVA pathway) and the 1-deoxy-D-xylose/2C-methyl-4-phospho-4D-erythritol pathway (DOXP/MEP pathway). The main difference between these two pathways lies in the different synthesis mechanisms and metabolic end products formed by the precursor of terpene synthesis, isopentenyl diphosphate (IPP), and its isomer, dimethylallyl diphosphate (DMAPP) [87]. In addition, a higher glycosidase activity can be maintained in non-*Saccharomyces cerevisiae* than in enological yeasts [88] because the terpenes bound to aminoglycosides can be directly released by the β-glucosidase hydrolysis of glycosidic bonds, while the hydrolysis of disaccharides and triglycerides also requires the participation of another glycosidase corresponding to the sugar group, such as α-L-mannosidase, α-L-apigenosidase, or α-L-arabinoside. In addition, the acid-catalyzed hydrolysis of the glycosidic terpene can be promoted, but the degree and rate of terpene release that this induces are not as great as those of the enzyme-catalyzed process [89].

### 6.3. Formation of Alcohols

Fermentation produces alcohols—typically ethanol, glycerol, higher alcohols, etc.—that impart fruit-like aromas to the wine, and they significantly contribute to the aromatic character of the finished product. Aromatic amino acids are metabolized by yeast via the Ehrlich pathway, and depending on the redox state of the cells, they can be further metabolized into their corresponding aromatic alcohols, indole 3-ethanol (chromanol), phenyl ethanol, and tyrosol, or oxidized to their corresponding acids, indoleacetic acid, phenylacetic acid, and 4-hydroxyphenyl acetic acid. These higher alcohols influence the aroma of the wine; this is especially true for 2-phenyl ethanol, which induces a strong rose aroma, is very popular in wines, and can generate the corresponding esters [90].

## 7. Artificial Modification of Aroma-Producing Yeast Based on Synthetic Biology

In addition to their ability to ferment and produce ethanol for various wines and alcoholic beverages, yeast cells have stable properties during fermentation, are easy to manipulate and scale up, grow faster compared to plants, and are more efficient in generating target products. Together with other advantages, such as having an extensive gene pool, with the development of synthetic biology, yeast cells are used more commonly as cell factories for the production of pharmaceuticals [91,92,93], biofuels [94], antibiotics [95], probiotics, flavorings, and many other high-value products. Some synthetic biological applications of yeast are shown in Table 2. *Yarrowia lipolytica* is a novel platform microorganism used in flavor and fragrance production; this food-grade yeast has the advantages of heat resistance and rapid growth, enabling it to act as a cell factory for the synthetic biological production of flavors and fragrances. Introducing new biosynthetic pathways into yeast cells can enable high-throughput production [96]. Arnesen et al. [97] constructed a platform bacteria for terpene production from *Yarrowia lipolytica*; 3-Hydroxy-3-methylglutaryl coenzyme A reductase (HMG), Mevalonate kinase (ERG12), ATP citrate lyase 1 (ACL), Enteric Salmonella acetyl-CoA synthetase (SEACs), Isopentyl diphosphate isomerase (IDI), ERG20F88W–N119W and the exchange natural Squalene promoter (PERG11-SQS) were constructed as monoterpene production platform strains. In addition, sesquiterpene, triterpene, and diterpene platform strains were obtained after different modification and treatment processes; these can be used for terpene biosynthesis, and provide a basis for the efficient construction of a cell factory that can be used in the production of various terpenoids. Agrawal et al. [98] first screened a type of geraniol synthase (tCrGES) in *Y. lipolytic* and then overexpressed different MVA pathway genes to enhance GPP precursors and increase the copy number of tCrGES to increase geraniol production. Finally, a simple metabolic modification was performed using lipolytic yeast to obtain a titer of 1 g/L, which was higher than the previous production of geraniol in *E. coli* and *Saccharomyces cerevisiae* (525 mg/L), reaching the highest level currently produced. Sandalwood and agarwood essential oils have long been used in perfumery and aromatherapy, but because of their low levels in plants and the difficulty of extraction, the development of efficient production methods has been challenging. Promdonkoy et al. [99] constructed the highly expressed strain FPPY005-39850 with all eight genes in the mevalonate pathway, and after combination with seven different terpene synthases screened from agarwood, sandalwood, and other closely related plants, the final engineered strain produced up to 101.7 ± 6.9 mg/L of aromatic terpenoids. Similarly, in the production of rose oil, a yeast strain overexpressing the MVA pathway was constructed to reconstruct the biosynthetic pathway of citronellol, replace the natural promoter ERG20 with the promoter of ERG7 (to enhance the expression of the geraniol synthesis module), and reconstitute the synthetic pathway of the three rose oil monoterpenes in this strain, which eventually led to a substantial increase in the yield of rose oil [100]. The production of terpene fragrances and aromatics using microbial cell factories requires only a change in the fermentation strain, and the resulting sesquiterpene hydrocarbons may represent a more sustainable feedstock for use in the aroma industry, and perhaps even replace traditional plant-based feedstocks in the future [101]. However, a variety of cell factories constructed using *Saccharomyces cerevisiae* are currently in use, and non-*Saccharomyces cerevisiae* has advantages such as ethanol tolerance, heat and inhibitor tolerance, and genetic diversity, which synthetic biology can extend [94]. In the future, the establishment of cell factories using non-*Saccharomyces cerevisiae* will be a research priority.

## 8. Application of Aroma-Producing Yeast

### 8.1. The Application of Essence

Flavor substances have pleasant aromas and are synthesized using a variety of spices, which have the characteristics of a wide available variety, specialization, and less material required. They are widely used in food, beverages, cosmetics, medical products, etc. With the development of metabolic engineering, genetic engineering, strain selection, and other technologies, it was found that a variety of the metabolites produced in fermented foods, such as volatile alcohols, esters, carbonyl compounds, terpenes, etc., have a strong natural flavor, and yeast, in particular, has a stronger flavor production capacity than bacteria and molds [108]. The use of biosynthetic techniques for the production of flavors has recently become more popular than chemical synthesis [109]. Terpenoids usually provide a wide variety of pleasant aromas, ranging from floral to fruity, woody, or balsamic notes, and for this reason, they constitute a very important class of compounds used in the flavor and fragrance industry. One of the most widely studied precursors used in the production of biotechnological monoterpenes is limonene, which has a pleasant orange- or citrus-like odor. It is produced by a key enzyme, limonene synthase (LS), after the catalyzation of the intramolecular cyclization of geranyl pyrophosphate (GPP) [110], which can also yield a variety of value-added aroma compounds, such as carvone, carvacrol, menthol, and pinene, among others [111]. In yeast cells, 2-phenyl ethanol, which is a naturally aromatic flavored alcohol with a rose fragrance and is widely used in the cosmetics, flavor, and food industries, is catabolized from L-phenylalanine, mainly through the Ehrlich pathway [112] and via biocatalysis catalyzed by non-conventional yeast (NCY) [113]. In addition, lactones are known for the various fruit flavors they impart, such as peach, pineapple, and mango, among which γ-decalactone (GDL) can be obtained by the β-oxidation of long-chain hydroxy fatty acids in yeast cells; γ-decalactone is commonly used in the cosmetics and fragrance industries [114]. Yeast extracts from brewer’s yeast contain a variety of amino acids (glutamic acid, glycine, and alanine) and are more flavorful than ordinary MSG (single glutamic acid), which opens up the opportunity to use modern biochemical techniques to make new food enhancers that combine nutrition and flavoring [115].

### 8.2. Application in the Food Industry

Aroma-producing yeasts are widely used in fermented foods, with fermented wine, vinegar, soy sauce, and bread as the main products. Three strains of aroma-producing yeast isolated from high-salt liquid fermentation mash greatly increase the concentrations of ester and alcohol flavor compounds in fermented sauces, and thus have promising applications in the brewing of soy sauce [44]. In fermented meats, yeasts can stabilize the typical red coloration, and improve the organoleptic properties of the meats via their deoxygenation capacity [116], while also inhibiting lipid oxidation, enabling the formation of esters and thus imparting intense aromas on fermented sausages when using high protein hydrolysis and high branched-chain alcohol production [117]. Yeast also plays a catalytic role in the fermentation process, and the addition of yeast induces an acceleration in sugar consumption and ethanol formation compared to spontaneous fermentation, resulting in a better color and richer aroma composition after inoculation with *Pichia kudriavzevii* during cocoa bean fermentation [118]. The improved process, including inoculation with the aroma-producing yeast HN006, can produce lily rice wines with higher contents of esters, free fatty acids, alcohols, aldehydes, ketones, olefins, volatile phenols, and thiazoles, and the aroma of the lily wine is also improved [119]. In addition, during the fermentation of pasta products, the use of a mixture of raw aromatic yeast and bacteria can change the aromatic and sensory properties of bread, especially under low-temperature conditions and during long fermentation, as more aroma components are produced and there are also increases in the numbers of furans, pyrazines, and volatiles derived from amino acids following protein hydrolysis and amino acid formation [120,121].

### 8.3. Application in Cosmetics

During thermal treatment, the lysis of yeast cells causes the release of a large number of endogenous enzymes that act on cellular components to produce flavonoids, sugars, vitamins, small peptides, amino acids, etc., which come to play an increasingly important role in the cosmetic industry. In a study by Dhurat [122], the use of yeast extracts in the treatment of pruritic symptoms yielded significantly faster and better results than using CO-containing lotions (significance, *p* = 0.0001). Yeast extracts have also shown effectiveness in cosmetics. Torula yeast extract and Torula yeast-derived glucose ceramide (GlcCer) increase dermal fibroblast proliferation and collagen production, as well as promote collagen gel contraction, thereby contributing to the maintenance of dermal elasticity and improving skin anti-wrinkling [123]. In addition, *Rhodotorula toruloides* is a non-brewing yeast with a natural carotenoid pathway, also known as an oleaginous yeast, which can accumulate high levels of lipids. R. Palmitic acid (PA, C16:0) and oleic acid (OA, C18:1) of *toruloides* Δ ^9^ and linoleic acid (LA, C18:2 Δ ^9, 12^) high in content, similar to the fatty acid composition of the stratum corneum. Kim et al. [124] used keratinocytes as a target and applied *R. toruloides* lipid extract to keratinocytes, resulting in a significant increase in the mRNA expression level of silk fibroin (silk fibroin can produce natural moisturizing factors, which can help maintain the moisture content of the stratum corneum, which is a key factor in skin hydration), and a dose-dependent relationship, reaching 2.83 times. The experimental results showed that *R. Toruloides* lipid extract can improve skin hydration by upregulating the expression of polyfilament protein, becoming an effective substrate for the production of cosmetics. Mannosylerythritol lipid (MEL), a glycolipid biosurfactant produced in large quantities by a variety of stretcher-like yeasts, has excellent interfacial properties and biochemical effects. For example, yeast glycolipid biosurfactants have moisturizing effects on dry skin, repairing effects on damaged hairs, and antioxidant and protective effects on skin cells, and therefore are gradually becoming used more widely in cosmetics [125].

## 9. Prospects

During fermentation, aroma-producing yeasts act on raw materials to produce a large number of volatile compounds with positive effects on the wine’s body. They are thus gradually becoming more widely used. Advanced alcohols, esters, and terpenes can impart floral and fruit aromas on wine, but some undesirable odor-producing substances can also be produced during the fermentation process, such as sulfur compounds [126] (SO_2_—rotten egg smell; methyl mercaptan—cooked cabbage smell; dimethyl trisulfide—garlic smell, etc.), and volatile acids (bitterness) can be produced under low-nitrogen conditions, etc. These undesirable odors account for a large proportion of fermented wine and can be detrimental to the desired qualities. The production of undesirable odors can be controlled by selecting appropriate yeast strains and adding appropriate nutrients during fermentation [127]. However, the number and variety of aroma-producing yeasts that were studied in China are small, and their aroma-producing functions have not been fully explored. Moreover, China occupies a wide geographical area and is rich in various types of yeast, so we should seek to develop our understanding of indigenous aroma-producing yeasts [128], form commercial strains, and apply them in relevant areas such as low-alcohol beverage, wine, soy sauce, fermented pasta, and condiment production [129], which will be beneficial to the development of China’s food fermentation industry. In addition, the application of emerging technologies, such as genetic engineering, protein engineering, cell engineering, and biosynthesis to genetically modify or alter the protein structures of aroma-producing yeasts will strengthen their aromatic qualities and resistance to unfavorable environmental factors, allowing their survival ability stronger. In addition to exploring the interactions and reaction mechanisms between raw aromatic yeasts and other microorganisms in the fermentation process, it will be beneficial to learn how to precisely control the fermentation conditions to increase fermentation efficiency, thus laying the foundation for the further development of the fermentation industry.

## Figures and Tables

**Figure 1 foods-12-03501-f001:**
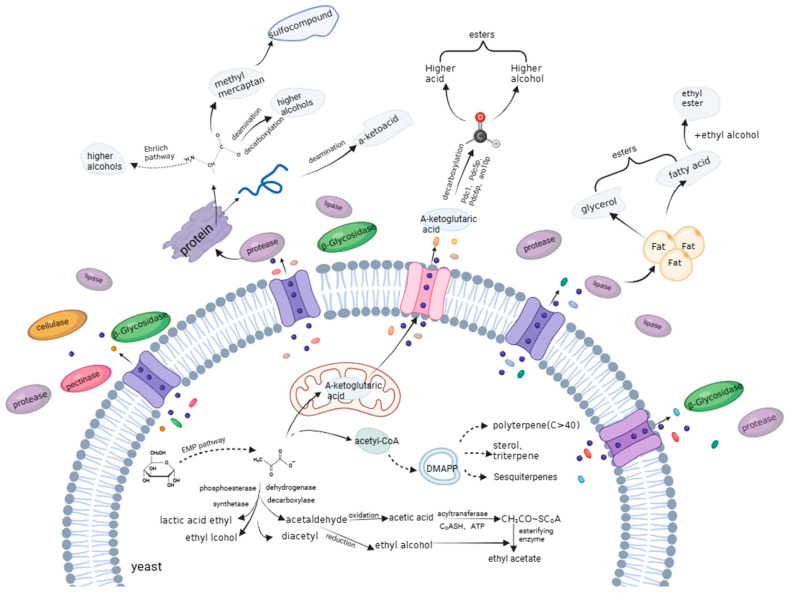
Aroma production mechanism of aroma-producing yeast.

**Table 1 foods-12-03501-t001:** Sources of aroma-producing yeasts.

Yeast	Source	Flavoring Substance	Aroma Characteristics	Main Applications	Literature
*Pichia jadinii*, *Torulaspora delbrueckii*, and *Kluyveromyces lactis*	Agricultural Microbial Culture Collection (CCMA)	Phenethyl acetate, etc.	Honey aroma	honey wine	Van [17]
*Torulaspora delbrueckii*, *Saccharomyces cerevisiae* and *Saccharomyces bayanus*	the Department of Microbiology collection; Grape spontaneous fermentation juice	2-phenylethanol, diethyl succinate, 2,3-butanediol, etc.	Rose fragrance		Sottil [18]
*T.delbrueckii* YH178 and YH179	Honeybee guts	-----	Sweet and grassy fragrance		Barry [19]
*Pichia*, *Candida*, and *Issatchenkia*	Indigenous yeast community	Terpenes and ethyl esters	Flowery and fruity aromas	plum wine	Chen [20]
*Candida glabrata*	Wine-producing area	Ethanol, isoamyl alcohol, ethyl nonanoic linalool, β-damascenone, terpene compounds, as well as phenyl acetate, ethyl benzoate, etc.	Rose aroma, tropical fruit aroma	wine	Han [21]
*Hanseniaspora opuntiae*, *Hansenula uvarum*, and *Hanseniaspora opuntiae*	Citrus wine and citrus orchard	Phenylethyl alcohol, 1-pentanol, ethyl acetate, isoamyl acetate, and phenethyl acetate; ethyl hexanoate, ethyl octanoate, and ethyl decanoate	Honey, rose, taste, pineapple, pear, and floral scent	orange wine	Hu [22]
*W. versatilis*, *C. sorbosivorans*, and *S. etchellsii*	Sauce residue	Ethyl esters and alcohols, furanone and maltol, pyrazine and phenyl ethyl alcohol	Fruity, mellow, sweet, and caramel aromas	Fermentation of sauces and wine	Wang [23]
*Wickerhamomyces anomalus* Y13, *Saccharomycopsis fibuligera* Y18, and *Torulaspora delbrueckii* Y22	Dumplings	2-pentylfuran, ethanol, hexanal and 1-hexanol, ethyl acetate, 2-heptanone and 2-nonanone, hexyl formate	Pineapple, varnish, balsamic vinegar; sweet fruit flavor, flower flavor, fruit flavor, peach flavor, aromatic fruit flavor	Flour product fermentation	Li [24]
*Meyerozyma guilliermondii*	Laobaigan Jiuqu	Higher fatty acid/acid ester, ethyl palmitate, ethyl linoleate	Fruity, sweet, rose fragrance	Laobaigan flavor liquor	Fu [25]
*Debaryomyceshansenii*	Fu brick tea	---	---	Fu brick tea fermentation	Xu [26]
*Zygosaccharomyces bailii* and *Pichia kudriavzevii*	Liquor fermented grains	Ethyl acetate and 3-methyl butyl acetate	---	wine	Li [27]

**Table 2 foods-12-03501-t002:** Synthetic biology of yeast.

Strain	Generating Substances	Key Pathways/Genes/Enzymes	Literature
*Saccharomyces cerevisiae*	Farnesene	strain improvement steps including mutagenesis, optimization of native metabolism, and enzyme engineering; overexpressed ADA, PK, PTA, and NADH-HMGr, and deleted RHR2 to generate AMR-5.	Meadows [102]
*Saccharomyces cerevisiae*	2-phenylethanol	overexpression of genes for catalytic enzymes, Aro9 and Aro10, and Aro80 transcription factor.	Kim [103]
*Saccharomyces cerevisiae*	β-ionone	β-carotene biosynthesis genes (crtI, crtE, and crtYB); carotenoid-cleavage dioxygenase from raspberry (RiCCD1)	Beekwilder [104]
*Saccharomyces cerevisiae*	Santalols	Replace its innate promotor with PHXT1; the genes related to santalol biosynthesis were overexpressed under the control of GAL promotors; GAL4 (a transcriptional activator of GAL promotors) and PGM2 (a yeast phosphogluco-mutase) were overexpressed.	Zha [105]
*Schizosaccharomyces pombe*	Vanillin	3-dehydroshikimate dehydratase, an aromatic carboxylic acid reductase (ACAR), and an *O*-methyltransferase; Knockout of the host alcohol dehydrogenase *ADH6*	Hansen [106]
*Pichia pastoris*	sesquiterpenoid (+)-nootkatone	overexpression of a *P. pastoris* alcohol dehydrogenase and truncated hydroxy-methylglutaryl-CoA reductase (tHmg1p).	Wriessnegger [107]

## Data Availability

The data used to support the findings of this study can be made available by the corresponding author upon request.

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
