# Peer review of "Reviewing the Source, Physiological Characteristics, and Aroma Production Mechanisms of Aroma-Producing Yeasts"

_foods, 2023, doi:10.3390/foods12183501_

Round 1

Reviewer 1 Report (New Reviewer)

The manuscript entitled "Review the source, physiological characteristics, and aroma production mechanism of aroma-producing yeast" was reviewed.  The topic of this study is interesting and fits well with the special issue. In this review study, the authors have attempted to introduce the source of aroma-producing yeasts, and review the mechanisms by which they generate aroma compounds. It has been written understandable; however it needs extensive English improvement. In overall, the authors are recommended to provide more schematic figures and biochemical routs, and also review genetic manipulations for improving physiological characteristics of such yeast species. Moreover, some modifications are recommended for authors that should be considered.  

TITLE

It seems that it is better to write "yeasts" instead of "yeast"

ABSTRACT

L8-9: The two first sentences can be merged.

Keywords: "aromatic mechanism" is not correct.

INTRODUCTION

Q: Introduction has been started with specific yeast "Saccharomyces". It is recommended that, in brief, the place of yeasts among other microorganisms in "three domain system of classification" and its taxonomy is mentioned with information on the numbers of yeast species, in particular those applied in food and cosmetic industries.

*Minor comments 

L-35-36: What yeast do you mean?

L-47: styles? Please replace it with a better word

L-61. Rewrite the sentence

L-67. "results show" should be replaced with "results showed"

Text

*The main comments:

Q: Please mention other substrates (other than methanol) used by Pichia for aroma generation.  

Q: What is the main substrate metabolized by Hansenula to generate aromatic compounds? Please mention in detail.

Q: "Artificial modification of aroma-producing yeast based on synthetic biology": A more comprehensive review in this section is recommended.

Q: The authors are expected to mention the main limitations and challenges for the development of application of yeasts in different industries discussed in this study.

Q: Figure 1 has not been referred in the text. The more important: If it has been extracted from other references, please mention it.  

*Minor comments

L-79-81: Please rewrite the sentences.

Table 1: Title should be corrected: yeasts

Table 1: Please check the journal instruction for reference style (Literature)

L-91: "The genus of Pichia" can be changed to "Pichia". The same modification is suggested for the next sections in the text.

L-104: What do you mean 'Hansen's yeast"?

L-112: "the strain can" should be changed to "the strain could"

L-129: here?

L-138-143: The sentence can be broken down to shorter sentences.  

L-415: "The Application of in Flavor" is not correct.

It has been written understandable, but needs extensive English polish.

Author Response

请参阅附件。

Reviewer 2 Report (New Reviewer)

The manuscript shows a full survey about a wide range of yeast species that are used in food production, concentrating on relevant flavour compounds.

Some details to be changed:

TITLE: as the manuscript deals mostly with typical Asian food and drinks this fact should be mentioned already in the title

in general: Thiols and the release of flavour-active compounds are missing and should be mentioned!

line 14 and elsewhere: as there are more enzymes involved in terpene production and release it should read: glycosidases

35: this sentence doesn't make sense, should be deleted

131: should read:... claussenii

152: ... trait called...

297: delete Burpho

347ff: release of bound terpenes to be mentioned

345: Chinese symbols must be replaced

Fig. 1: glycosidases!

415: correct typing needed

see above

Author Response

Reviewer 3 Report (New Reviewer)

I think this is a well written paper with good application awareness.

Author Response

请参阅附件。

Round 2

Reviewer 1 Report (New Reviewer)

The authors have responded to the comments, polished the English and improved the manuscript quality. It can be accepted for publication in FOODS.

The quality of English language has been improved, however it need minor polishing.

Author Response

     Thank you for reviewing this chapter amidst your busy schedule. I am delighted to see your affirmation of the article. Your affirmation is an excellent encouragement for me. In the coming days, I will work harder and study harder. Thank you again for your review and affirmation! Wishing you all the best.

Best regards,

All authors.

This manuscript is a resubmission of an earlier submission. The following is a list of the peer review reports and author responses from that submission.

Round 1

Reviewer 1 Report

Lines 38-39: the concept that you are trying to introduce in this review of aroma-producing and non-producing yeasts does not seem to me to be correct.

The secondary metabolism of Saccharomyces and non-Saccharomyces yeasts share many pathways. The production in many of these metabolites and the concentrations they reach at the end of fermentation also depend on the matrix where the yeast is developing.

In the introduction this sentence says:

“non-Saccharomyces yeasts can be used in fermentation to produce specific aromas, which bring special qualities to the wine after fermentation, and non-Saccharomyces yeasts are gradually emerging as a source of aromatic compounds in wine[3,4]. These are also known as aroma-producing yeast. Aroma producing yeast is a class of strains that can produce alcohols, aldehydes, organic acids, and furan compounds with aromatic metabolites through fermentation.”

So it follows that the review is going to be about non-Saccharomyces, but the end of the review details modeificationtalks about genetically modified Saccharomyces yeasts.

Throughout the manuscript there are several difficult to understand or erroneous phrases.

Line 58: acetic acid? He meant to put acetates'.

Lines 157 "Zygosaccharomyces rouxii is responsible for the conversion of high concentrations of sugar to ethanol, and is more fructophilic than glucose."  Glucophilic?

“Cai[46] et al. used 166 Candida for aroma enhancement in the brewing of a new type of biopeptide yellow wine, and the results show that the addition of pseudo filament yeast culture at 2-3% during  compound fermentation could significantly enhance the aroma of yellow wine; the wine also had a more mellow taste and better overall quality”. It is difficult to understan the “biopeptide yellow wine”

“A variety of glycosidases in aroma-producing yeast hydro lyzeglycosidic bonds to release large amounts of volatile terpenes, as well as acetyl-CoA and pyruvic acid to release monoterpenes (C10), sesquiterpenes (C15), and diterpenes (C20), via the MVA pathway and DOXP/MEP pathway, respectively[81]. The alcohol and malic acid β-glucosidase activity of yeast that is activated during malolactic fermentation increases the free terpene content[82], and these terpenes play a major role in the formation of wine aromas”.  This is wrong

All yeast names should be written in italics both Saccharomyces cerevisiae and the rest of the non-Saccharomyces yeast names. The manuscript contains a large number of such errors and many different types of font formatting throughout the text.

Reviewer 2 Report

Thank you for the opportunity to read the manuscript of the work entitled: Review the source, physiological characteristics, and aroma production mechanism of aroma-producing yeast. This work is a form of a narrative description of the use of various yeast species for the production of odor or fragrance ingredients. The basic problem I noticed while reading the manuscript is that the work does not exhaust the indicated topic, has an undefined form and the manuscript itself is characterized by low attention to details.

The Authors do not use the correct form of writing of the full species name of the discussed organisms, there is no homogeneity in the use of Latin names in italics, in some fragments the names appear in a completely random font. The rules for using the species nomenclature of organisms indicate that the subsequent indication of this species allows the use of the generic abbreviation. This rule does not apply in this text.

Perhaps a more serious problem is the lack of indication of the working method. Such a narrative form of expression is of course valuable from an educational point of view, but nowadays it is more about systematic reviews. The tables used in this work show that the authors wanted to maintain (at least partially) such a systematic nature of the work. However, in this case, the exact criteria for inclusion in the review, time frames and other critical conditions should be indicated.

Mistakes:

- captions under graphics and table titles are incorrect and incomplete.

- names of chemical compounds are written once in uppercase and once in lowercase, so there is no homogeneity

- errors in the spelling of species names have been discussed above

- there are abbreviations in the text that have not been introduced

- the sentence in line 54 does not refer to the source 

- the authors incorrectly cite sources: the reference in square brackets should not be in superscript; Wang [11] et al. - is incorrect - should be: Wang et al.[11]. In the References chapter, the sources have not been written in accordance with the publisher's guidelines

- the work does not exhaust the topic, especially the chapter on the cosmetics industry is insufficient

- the work has an incorrect structure, even a conclusion is missing. I have already written about the lack of a chapter: material and method above.

- in the text there are numerous imprecise terms, e.g. beverage wine